# Manipulation of Energy Flow with X-Type Vortex

**Han Zhang [1], Tianhu Zhang [1] , Xinying Zhao [2] and Xiaoyan Pang [1],\***

[1]  School of Electronics and Information, Northwestern Polytechnical University, Xi'an 710072, China
[2]  School of Physics & Information Technology, Shaanxi Normal University, Xi'an 710061, China
\*  Correspondence: xypang@nwpu.edu.cn

**Abstract:** In this study, a new method for manipulating energy flow in a 3D vector field is proposed. In this method, an azimuthally-polarized beam with a noncanonical vortex, the X-type vortex, is focused in a high-numerical aperture system. It is found that, instead of the invariance of the energy flow which is characteristic of the traditional vortex (i.e., canonical vortex), both the longitudinal and the transverse energy flows in virtue of the X-type vortex rotate around the beam center as the beam propagates, and this rotational behavior (including the maxima location and the rotational angle) can be adjusted by the anisotropic parameter and the order the X-type vortex. Through defining a complex transverse Poynting field and applying the equivalence principle, the transverse energy flow and its topological reactions are discussed in the focal plane. Our result shows that, by changing the anisotropic parameter of the X-type vortex, rich topological reactions will occur, resulting in various distribution patterns of the energy flow, such as multi vortex-type singularities around the beam center. Our research demonstrates newly-observed features of the X-type vortex and also provides a simple method to manipulate energy flows both along longitudinal and transverse directions, which will be useful in optical manipulations.

**Keywords:** Poynting vector; optical vortex; singularity; energy flow

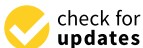



## 1. Introduction

The energy flow of light has been studied for more than 100 years since the possibility of backward energy was revealed in the near-focus field in 1919 [1]. Later, in 1959, the energy flow was also analyzed near the Airy rings in a classical article [2]. Because of its important role in both fundamental and applied research, the study of energy flow has become more vigorous in recent decades [3–23]. In fundamental research, the energy flow provides a natural method for exploring the most intimate features of an optical field, such as the (intrinsic) energy flow being divided into a spin part and an orbital part corresponding to the two different angular momenta of an optical field [3–6], which reflects the physical nature of light; the behaviors of the energy flow can connect to the topological reactions of optical singularities, which supports a method of explaining the special features of singular optics [7,8]. On the other hand, energy flow has been utilized in optical manipulations. The absorptive particle can move along the direction of the energy flow and the velocity of the movement is proportional to the modulus of the energy flow [9,10]. Recently, beams with backward energy flow have also attracted a lot of interest for their role as a 'tractor' to exert pulling force on a particle in manipulation schemes [11–16]. Many methods have been proposed in manipulating energy flows, such as tailoring the phase structures [6,17,18] and/or polarization distributions [19–22]. Research on energy flow continues to be expanded on in fundamental and applied optics [16,23].

An optical vortex usually refers to the canonical vortex with a constant phase gradient around its center [8,24–26]. Since it has peculiar characteristics, such as carrying orbital angular momentum, the optical vortex has been studied extensively and utilized in a wide range of applications, such as in optical tweezers [27], optical communications [28], imaging [29], microscopy [30,31], etc. The vortex also plays a key role in most research on

the manipulation of energy flows [12,14–20,22]. Besides the well-known canonical vortex, there also exists the noncanonical vortex, which has not received as much attention in most studies [32–36]. For a noncanonical vortex, the phase gradient is no longer constant along the azimuthal direction; thus, there will exist an 'anisotropic parameter' characterizing the phase distribution, which actually provides more freedom for beam structure [33,36]. Very recently, an X-type vortex, as one type of the noncanonical vortex, was proposed, and it was found that this noncanonical vortex could shape the intensity distribution in rich structures in 3D vector fields [36]. In this article, we will use the X-type vortex and show its effects on energy flows.

## 2. Materials and Methods

The X-type vortex is a type of noncanonical optical vortex, with an anisotropic phase distribution [36]. The transverse field of an optical beam embedded with an X-type vortex can be expressed as:

$$V^{(X)}(x,y) = A(x,y)\left(\sqrt{x^2+y^2}\right)^l \frac{(x+i\sigma_c y)^l}{\left(\sqrt{x^2+\sigma_c^2 y^2}\right)^l}, \tag{1}$$

where $\sigma_c$ is the anisotropic parameter determining the phase distribution and $l$ ($l \in N$) represents the order of the vortex. When $\sigma_c = \pm 1$, the X-type vortex degenerates into a conventional (canonical) vortex. It is also convenient to re-write the Equation (1) in the polar coordinates ($x = \rho cos\phi, y = \rho sin\phi$):

$$V^{(X)}(\rho,\phi) = A(\rho,\phi)\rho^l e^{il\arctan(\sigma_c \tan\phi)}, \tag{2}$$

Which shows that the phase is a nonlinear function of azimuthal direction $\phi$ and the phase gradient along the $\phi$ direction equals $l\sigma_c / \left(\cos^2\phi + \sigma_c^2 \sin^2\phi\right)$. Examples of the phase distributions are depicted in Figure 1a, where the plot with $l = 1$ and $\sigma_c = 1$ represents the conventional (canonical) vortex with order 1, and one can see that the phase changes uniformly along the $\phi$ direction. The phases in the plots with $\sigma_c > 1$ change faster near the $x$-axis, where the phase in the plot with $\sigma_c < 1$ changes faster near the $y$-axis.

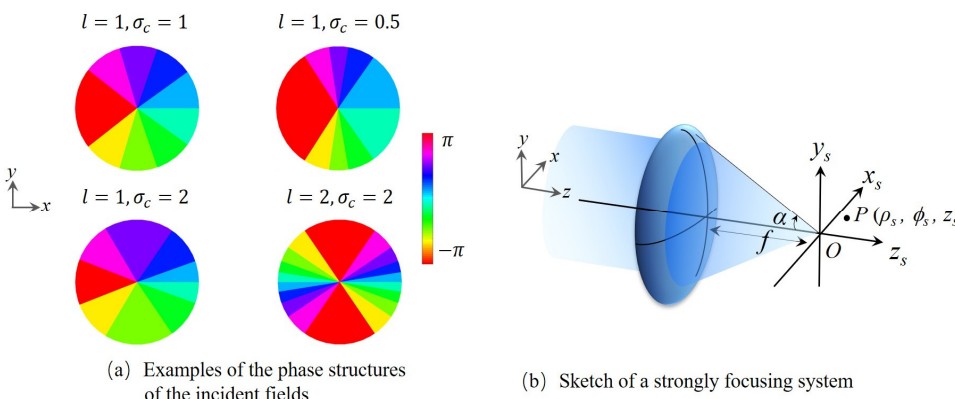

(a) Examples of the phase structures of the incident fields

(b) Sketch of a strongly focusing system

**Figure 1.** A strongly focusing system with an X-type vortex beam as the incident field.

Next, we will use this X-type vortex to construct a 3D vector field in a high numerical aperture (NA) system. First, consider a high NA system with a semi-aperture angle $\alpha$ and a focal length $f$. The focus of this system is located at point O, the origin of the Cartesian coordinate system, see Figure 1b. Then, assume that a Gaussian beam embedded with an X-type vortex is incident upon this focusing system, i.e., the complex amplitude of the incident field can be expressed by Equation (2) with $A(\rho,\phi) = exp\left(-\frac{\rho^2}{w_0^2}\right)$, where $w_0$ is the waist size. This X-type vortex can be generated by using a phase-spatial light modulator or a programmable q-plate [37–39]. According to the Richards-Wolf vector diffraction theory, the electric/magnetic field at a point $P(\rho_s, \phi_s, z_s)$ in the focal region can be written as [2]

$$\mathbf{U}(\rho_s, \phi_s, z_s) = -\frac{ikf}{2\pi} \int\limits_0^\alpha \int\limits_0^{2\pi} V^{(X)}(\rho, \phi) \, \mathbf{Q}_{E(H)}(\theta, \phi) \sqrt{\cos\theta} e^{ikz_s \cos\theta}$$
$$\times e^{ik\rho_s \sin\theta \cos(\phi-\phi_s)} \sin\theta d\theta d\phi, \tag{3}$$

where $\mathbf{U}(\rho_s, \phi_s, z_s)$ represents the 3D electric field ($\mathbf{E}(\rho_s, \phi_s, z_s)$) or magnetic field ($\mathbf{H}(\rho_s, \phi_s, z_s)$), and where $k = 2\pi/\lambda$ is the wave number with $\lambda$ denoting the wavelength of the free space. $\mathbf{Q}_E(\theta, \phi)(\mathbf{Q}_H(\theta, \phi))$ is the polarization matrix of the electric (magnetic) field, which can be expressed as

$$\mathbf{Q}_{E(H)}(\theta, \phi) = \begin{bmatrix} \cos\theta + \sin^2\phi(1 - \cos\theta) \\ (\cos\theta - 1)\cos\phi\sin\phi \\ -\sin\theta\cos\phi \end{bmatrix} a(\theta, \phi) + \begin{bmatrix} (\cos\theta - 1)\cos\phi\sin\phi \\ \cos\theta + \cos^2\phi(1 - \cos\theta) \\ -\sin\theta\sin\phi \end{bmatrix} b(\theta, \phi), \tag{4}$$

where $a(\theta, \phi)$ and $b(\theta, \phi)$ are the weight functions for the $x$-polarized and $y$-polarized components of the incident beam, respectively. Assume this incident beam is also azimuthally polarized with order $m$; thus, for the electric field:

$$\begin{pmatrix} a(\theta, \phi) \\ b(\theta, \phi) \end{pmatrix} = \begin{pmatrix} -\sin m\phi \\ \cos m\phi \end{pmatrix}, \tag{5}$$

and for the magnetic field:

$$\begin{pmatrix} a(\theta, \phi) \\ b(\theta, \phi) \end{pmatrix} = \begin{pmatrix} -\cos m\phi \\ -\sin m\phi \end{pmatrix}. \tag{6}$$

For simplicity, from here on, we only consider the case of $m = 1$; the other cases can be derived in the same way. By substituting Equations (5) and (6) into Equation (4), we can obtain two simple expressions for the polarization matrix:

$$\mathbf{Q}_E(\theta, \phi) = \begin{bmatrix} -\sin\phi \\ \cos\phi \\ 0 \end{bmatrix}, \tag{7}$$

$$\mathbf{Q}_H(\theta, \phi) = \begin{bmatrix} -\cos\theta\cos\phi \\ -\cos\theta\sin\phi \\ \sin\theta \end{bmatrix}. \tag{8}$$

thus, the electric field $\mathbf{E}(\rho_s, \phi_s, z_s)$ and magnetic field $\mathbf{H}(\rho_s, \phi_s, z_s)$ in the focal region can be calculated as:

$$\mathbf{E}(\rho_s, \phi_s, z_s) = \begin{bmatrix} e_x \\ e_y \\ e_z \end{bmatrix} = -ik \int_0^\alpha L(\theta) \begin{bmatrix} I_x(\theta, \phi) \\ I_y(\theta, \phi) \\ 0 \end{bmatrix} d\theta, \tag{9}$$

$$\mathbf{H}(\rho_s, \phi_s, z_s) = \begin{bmatrix} h_x \\ h_y \\ h_z \end{bmatrix} = -ik \int_0^\alpha L(\theta) \begin{bmatrix} -\cos\theta \, I_y(\theta, \phi) \\ \cos\theta \, I_x(\theta, \phi) \\ \sin\theta \, I_z(\theta, \phi) \end{bmatrix} d\theta, \tag{10}$$

where

$$L(\theta) = \sqrt{\cos\theta} e^{-(f\sin\theta)^2/w_0^2} f^l (\sin\theta)^{l+1} e^{ikz_s \cos\theta}, \tag{11}$$

and

$$I_x(\theta, \phi) = \int_0^{2\pi} \frac{-\sin\phi(\cos\phi + i\sigma_c \sin\phi)^l}{\left(\sqrt{\cos^2\phi + \sigma_c^2 \sin^2\phi}\right)^l} e^{ik\rho_s \sin\theta \cos(\phi-\phi_s)} d\phi, \tag{12}$$

$$I_y(\theta, \phi) = \int_0^{2\pi} \frac{\cos\phi(\cos\phi + i\sigma_c \sin\phi)^l}{\left(\sqrt{\cos^2\phi + \sigma_c^2 \sin^2\phi}\right)^l} e^{ik\rho_s \sin\theta \cos(\phi-\phi_s)} d\phi, \tag{13}$$

$$I_z(\theta, \phi) = \int_0^{2\pi} \frac{(\cos\phi + i\sigma_c\sin\phi)^l}{\left(\sqrt{\cos^2\phi + \sigma_c^2\sin^2\phi}\right)^l} e^{ik\rho_s\sin\theta\cos(\phi-\phi_s)}\,\mathrm{d}\phi. \tag{14}$$

The energy flow, which is described by the time-averaged Poynting vector **P**, can now be written in terms of the expressions of these 3D electric and magnetic fields, as in [3,12]:

$$\mathbf{P} \propto \mathrm{Re}[\mathbf{E} \times \mathbf{H}^*] = \begin{pmatrix} p_x \\ p_y \\ p_z \end{pmatrix} = \begin{pmatrix} \mathrm{Re}[e_y h_z^*] \\ -\mathrm{Re}[e_x h_z^*] \\ \mathrm{Re}[e_x h_y^* - e_y h_x^*] \end{pmatrix}, \tag{15}$$

where $\mathrm{Re}[\cdot]$ means the real part and the superscript * denotes the complex conjugate. The following discussions on the energy flow are mainly based on the equations derived in this section.

## 3. Results and Discussions

In this section, we first discuss the behaviors of the energy flow as the beam propagates. After that, the properties of the transverse energy flow in the focal plane will be analyzed.

### 3.1. Longitudinal Energy Flow along the Propagation Direction

Let us first analyze the longitudinal energy flow, which is the longitudinal component of the Poynting vector $p_z$ along the propagation direction. When $\sigma_c = 1$, i.e., the case for the canonical vortex, the integral with azimuthal angle $\phi$ in Equations (12)–(14) can be calculated and the first kind of Bessel functions with trigonometric functions will be obtained. Since the relations of the transverse components of the electric field and the magnetic field (see Equations (9)–(13)), it can be calculated easily that $p_z$ has a circular symmetry, i.e., $p_z(\rho_s, \phi_s + \Delta_\phi, z_s) = p_z(\rho_s, \phi_s, z_s)$, with $\Delta_\phi$ being an arbitrary azimuthal angle. This means that the longitudinal energy flow is uniformly distributed along the azimuthal direction, which can be seen in Figure 2b. In addition, when $\sigma_c \neq 1$, the X-type vortex (noncanonical case), this symmetry will be broken, see Figure 2a,c.

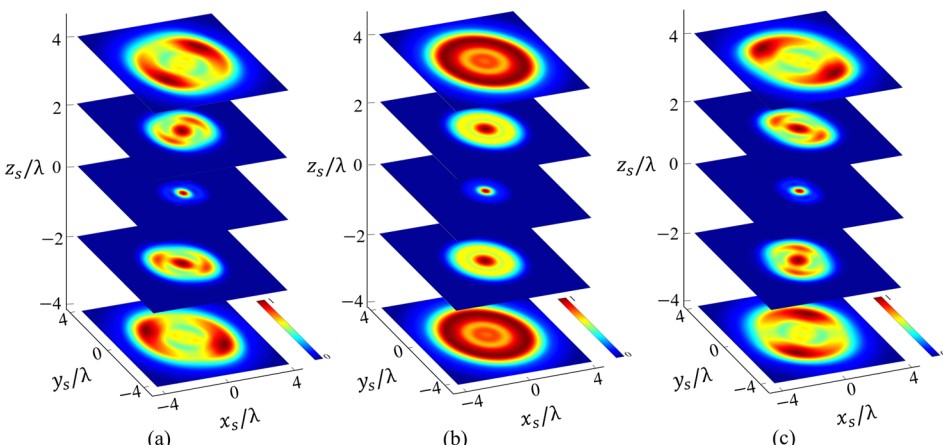

(a)　　　　　　　　　　　　(b)　　　　　　　　　　　　(c)

**Figure 2.** The longitudinal energy flow $p_z$ along the propagation direction: (**a**) $\sigma_c = 0.5$; (**b**) $\sigma_c = 1$; (**c**) $\sigma_c = 2$. In all plots $l = 1$.

Figure 2 depicts the distribution of the longitudinal component $p_z$ in different transverse planes along the propagation direction, where the order of the beam is chosen as $l = 1$ and the anisotropic parameter $\sigma_c = 0.5$ in plot (a), $\sigma_c = 1$ plot (b), and $\sigma_c = 2$ in plot (c). In this figure, the semi-aperture angle $\alpha$ is set as $60°$, and, unless otherwise specified, $\alpha = 60°$ in this article. In this figure, $p_z$ is always positive except certain points where $p_z = 0$, which implies that the longitudinal energy flow (if it exists) always points to the $+z_s$ direction. Through observing this figure, we can obtain the hypotheses that: (a) for any value of $\sigma_c$ (when $l = 1$), the maximum of the energy is located at the focus on the focal plane,

while, when the propagation distance is far from the focal plane, the maximum/maxima is/are gradually 'thrown out' of the beam center; (b) for $\sigma_c \neq 1$ (the X-type case), $P_z$ has two maxima in the transverse plane (which has a distance from the focal plane). Also, for $|\sigma_c| < 1$, the two maxima in the $-z_s$ space are more likely to stay in the second and fourth quadrants and, in the $+z_s$ space, to stay in the first and third quadrants. For $|\sigma_c| > 1$, the trend is just the opposite; c) more interestingly, when $\sigma_c \neq 1$, the distribution pattern of $P_z$ rotates in a counterclockwise manner along the beam propagation direction. This indicates that *the X-type vortex leads to a rotation of the longitudinal energy flow as the beam propagates*. In the following, this rotational behavior will be examined more deeply.

First of all, in order to characterize this rotational behavior quantitively, a rotational angle $\varphi_r$ is introduced, as shown in Figure 3. $\varphi_r$ is defined as the azimuthal angle of one maximum point of $p_z$. In this study, to remain consistent, this maximum point is always chosen as the one near the $+x_s$ axis from the $-z_s$ space. In other words, first, we fix one maximum point in the $-z_s$ space (the condition for this point is that it is near the $+x_s$ axis), then we follow the position of this point as the beam propagates and record its rotational angle $\varphi_r$. For instance, in Figure 3, the maximum point is the one in the fourth quadrant and $\varphi_r$ is about $-37°$. Thus, the variation in the rotational angle $\varphi_r$ with the beam propagation can be drawn, and, as shown in Figure 4, four curves of $\varphi_r$ for $\sigma_c = 0.1, 0.5, 2,$ and $10$ are depicted. In this figure, the range is $-14\lambda < z_s < 14\lambda$ and the order $l = 1$. From this figure, the rotational behavior and its counterclockwise manner of the longitudinal energy flow $p_z$ can be seen more clearly. The accumulated rotational angle (here denoted by $\int \varphi_r$) from $-14\lambda$ to $14\lambda$ is quite big, which is more than $100°$ in all four of these cases. Also, this rotation can be manipulated by the propagation distance $z_s$ and the anisotropic parameter $\sigma_c$. As $|z_s|$ increases and/or $\sigma_c$ moves further from 1, the rotational behavior becomes more obvious and the accumulated rotational angle $\int \varphi_r$ becomes bigger: $\int \varphi_r = 112.6°$ for $\sigma_c = 0.1$, and $10$ and $\int \varphi_r = 100.4°$ for $\sigma_c = 0.5, 2$. As $|z_s|$ approaches $+\infty$, $\int \varphi_r$ is infinitely nearer to $180°$. In addition, the curves for $\sigma_c = 0.1$ and $10$ are the same except for a constant difference, which is also the case for $\sigma_c = 0.5$ and $2$. This is because, in each group, these two values of $\sigma_c$ are reciprocal of each other, and, according to Equations (1) and (2), replacing $\sigma_c$ with $1/\sigma_c$ is equivalent to exchanging the $x$ and the $y$ coordinates. Furthermore, one also can find that the curves which are very near focus are very steep; this is mainly caused by the dramatic changing of the wavefront spacings in the high NA system [40,41].

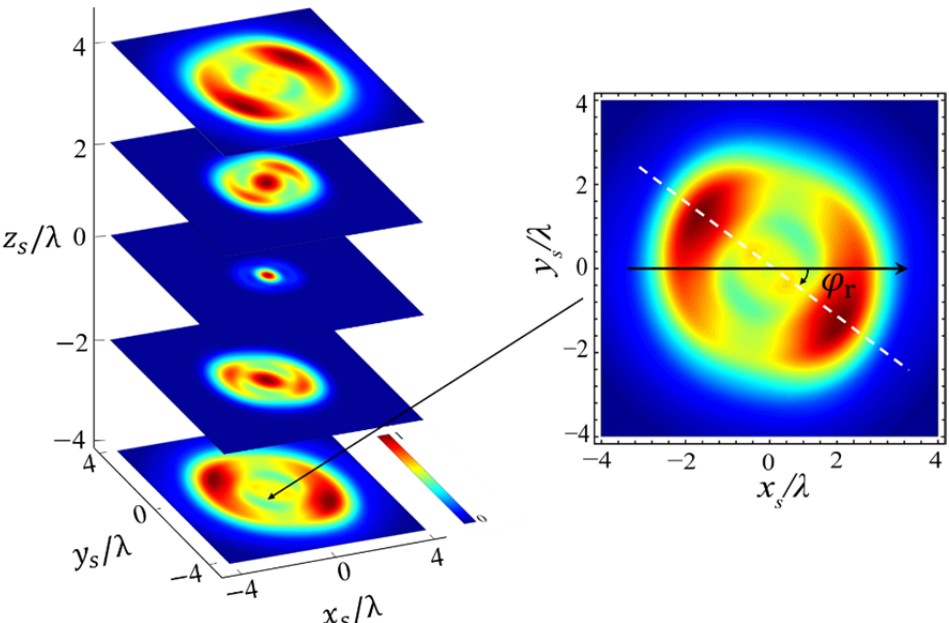

**Figure 3.** Definition of a rotational angle $\varphi_r$. Here, $\sigma_c = 0.5, l = 1, z_s = -4$.

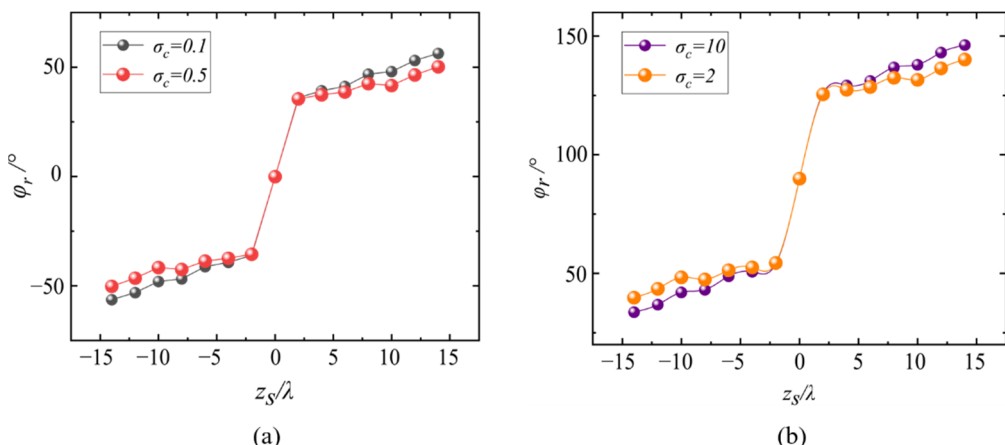

**Figure 4.** Variation in the rotational angle $\varphi_{\mathbf{r}}$ for the longitudinal energy flow $p_z$ with the beam propagation. Here, (**a**) $\sigma_c = 0.1$, 0.5; (**b**) $\sigma_c = 2$, 10. In both plots $l = 1$.

In the above analysis, the beam order $l$ is set as 1, and, here, we will show that changing the order $l$ can also manipulate the rotational behavior of $p_z$. The distribution of $p_z$ along the propagation direction for the X-type vortex with order 2 ($l = 2$) is illustrated in Figure 5. By observing Figure 5 with Figure 2, we can find: (a) for $l = 2$, the beam center is always hollow, i.e., the energy of $p_z$ is mainly distributed around the beam center, which is the main difference in the distribution of $p_z$ between $l = 2$ and $l = 1$. Further, more generally, for any $l \neq 1$, the distribution of $p_z$ always exhibits a hollow shape. This result can be obtained directly from the expression of $e_x$, $e_y$, and $P_z$, using Equations (9)–(15). Consider the simplest case $\sigma_c = 1$; from Equations (12) and (13) we can obtain $I_{x(y)} = C_{x(y)} J_{l\pm1}(k\rho_s \sin\theta)$, with $J_{l\pm1}$ being the first kind of Bessel function. This expression implies that, only when $l = 1$, the $e_x$ and $e_y$ (also $h_x$, and $h_y$) will not be zero along the $z_s$ axis ($\rho_s = 0$) (note in this article $l \in \mathbb{N}$). Therefore, $p_z (\rho_s = 0) = 0$ for any $l \neq 1$. Furthermore, since a property of the Bessel function is that, as $l$ increases, the radius of the hollow part widens, which can be observed from the comparison of Figures 3 and 5. Although these two results can be derived from the properties of the Bessel function, they can be easily generalized to the case of $\sigma_c \neq 1$. (b) The longitudinal component $p_z$ in $l = 2$ also has a similar two maximum points and rotational behaviors to those in $l = 1$. For $\sigma_c \neq 1$, $p_z$ has two main maxima in the transverse plane instead of maximum rings of $\sigma_c = 1$, and the positions of the maxima are located in the same quadrants as in in $l = 1$. More importantly, the distribution pattern of $p_z$ also rotates in a counterclockwise manner along the propagation direction, which is the same as it is in $l = 1$.

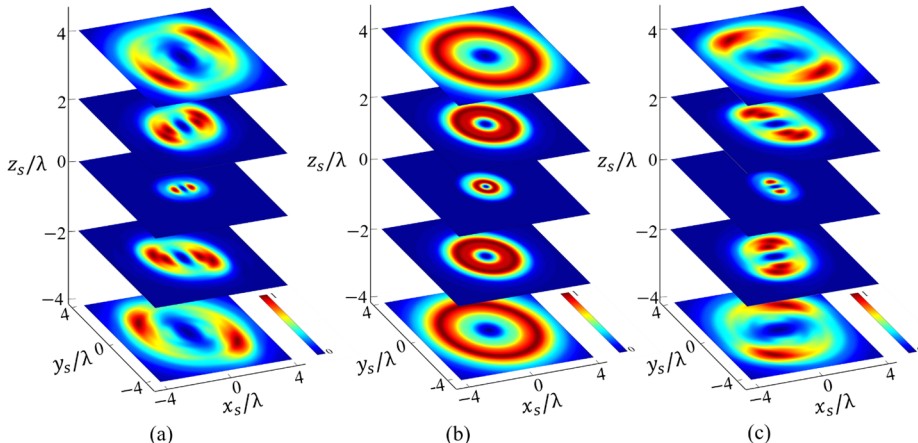

**Figure 5.** The longitudinal energy flow $p_z$ along the propagation direction: (**a**) $\sigma_c = 0.5$; (**b**) $\sigma_c = 1$; (**c**) $\sigma_c = 2$. In all the plots $l = 2$.

Similarly, the rotational angle $\varphi_r$ for $l = 2$ can also be drawn, as shown in Figure 6. By comparing Figure 6 with Figure 4, one can see that the rotation for $l = 2$ goes more smoothly as the beam propagates, and the accumulated rotational angle for $l = 2$ is generally bigger than its corresponding one for $l = 1$, i.e., $\int \varphi_r (l = 2) = 117.5° > \int \varphi_r (l = 1) = 112.6°$ for $\sigma_c = 0.1, 10$, and $\int \varphi_r (l = 2) = 108.9° > \int \varphi_r (l = 1) = 100.4°$ for $\sigma_c = 0.2, 5$. Also, it is not hard to calculate that, as $l$ increases, this accumulated rotational angle $\int \varphi_r$ becomes slightly bigger.

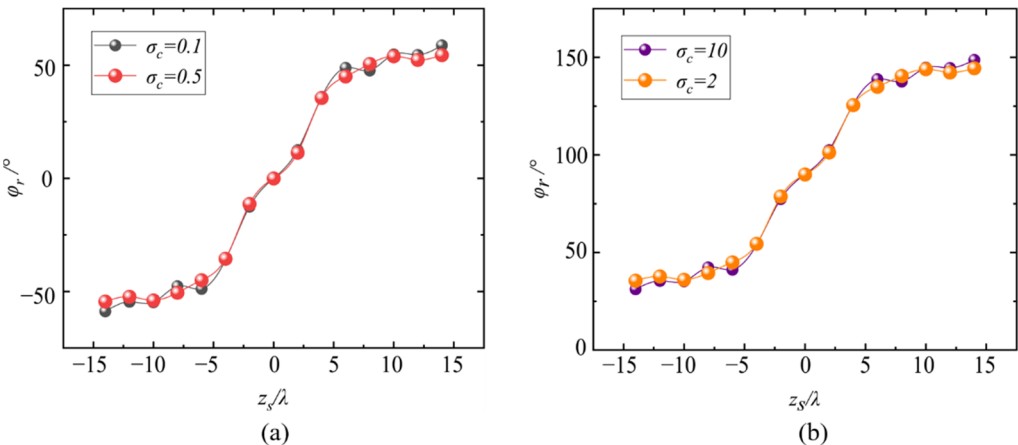

(a)

(b)

**Figure 6.** Variation in the rotational angle $\varphi_r$ for the longitudinal energy flow $p_z$ with the beam propagation. Here, (**a**) $\sigma_c = 0.1, 0.5$; (**b**) $\sigma_c = 2, 10$. In both plots $l = 2$.

### 3.2. Transverse Energy Flow along the Propagation Direction

Transverse energy flow is one of the characteristics of 3D vector optical fields. In addition to $p_z$, the transverse energy flow, $\mathbf{p}_{xy} = (p_x, p_y)$, as it will be shown, can also rotate with the beam propagating.

Figure 7 depicts the transverse energy flow $\mathbf{p}_{xy}$ in different transverse planes along the propagation direction. Here, the energy strength $\left| \mathbf{p}_{xy} \right|$ is presented by different colors and the direction of $\mathbf{p}_{xy}$; the flow lines are drawn by black lines with an arrow. It can be seen that, firstly, the distribution of the transverse energy flow $\mathbf{p}_{xy}$ for $\sigma_c \neq 1$ can also rotate along the propagation direction, and this rotation can be found to be counterclockwise. This indicates that *the X-type vortex can not only cause the rotation of the longitudinal flow $p_z$, but can also make the transverse energy rotate with the beam propagating*. Similarly, the rotation of the transverse energy flow can be measured quantitively by using the rotational angle $\varphi_r$, while $\varphi_r$, here is defined with respect to the transverse energy distribution.

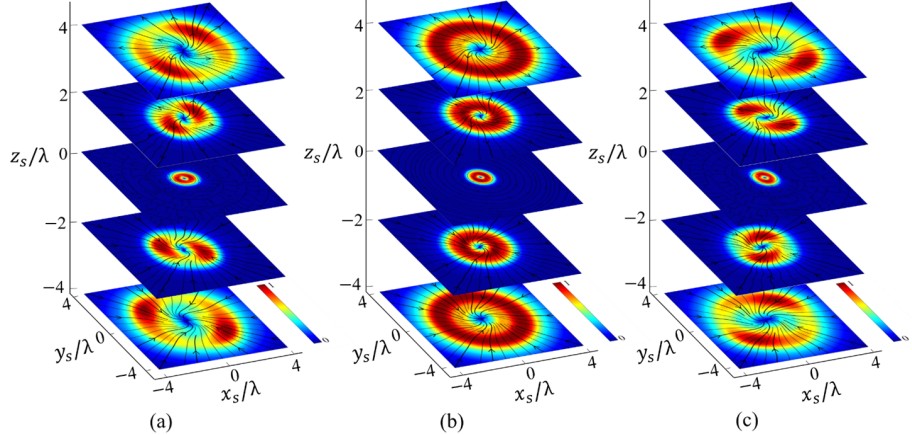

(a)

(b)

(c)

**Figure 7.** The transverse energy flow $\mathbf{p}_{xy}$ along the propagation direction: (**a**) $\sigma_c = 0.5$; (**b**) $\sigma_c = 1$; (**c**) $\sigma_c = 2$. In all the plots $l = 1$.

The variation in the rotational angle $\varphi_r$ for the transverse energy flow along the propagation direction is drawn in Figure 8. One can see that the accumulated rotational angle $\int \varphi_r$ is slightly bigger than its corresponding longitudinal component, with $\int \varphi_r = 119.9°$ for $\sigma_c = 0.1, 10$ and $\int \varphi_r = 116.6°$ for $\sigma_c = 0.5, 2$. It is worth noting that, although the overall rotational tendency is counterclockwise, in the range of about $|z| < \lambda$, $\mathbf{p}_{xy}$ will show a short clockwise rotation. Due to the limitation of the sampling points in Figure 7, this abnormal rotation cannot be seen. In Figure 9, the clockwise rotational behavior is shown, and one can find that, in the range $-0.8\lambda < z_s < 0.8\lambda$, the accumulated clockwise rotational angle is about $-22.6°$ for $\sigma_c = 0.1, 10$ and $-36.9°$ for $\sigma_c = 0.5, 2$.

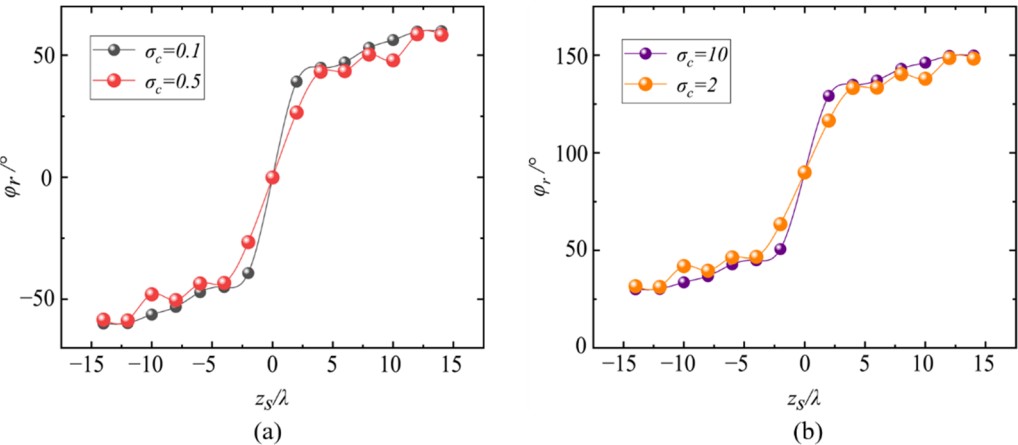

**Figure 8.** Variation in the rotational angle $\varphi_r$ for the transverse energy flow $\mathbf{p}_{xy}$ with the beam propagation. Here, (**a**) $\sigma_c = 0.1, 0.5$; (**b**) $\sigma_c = 2, 10$. In both plots $l = 1$.

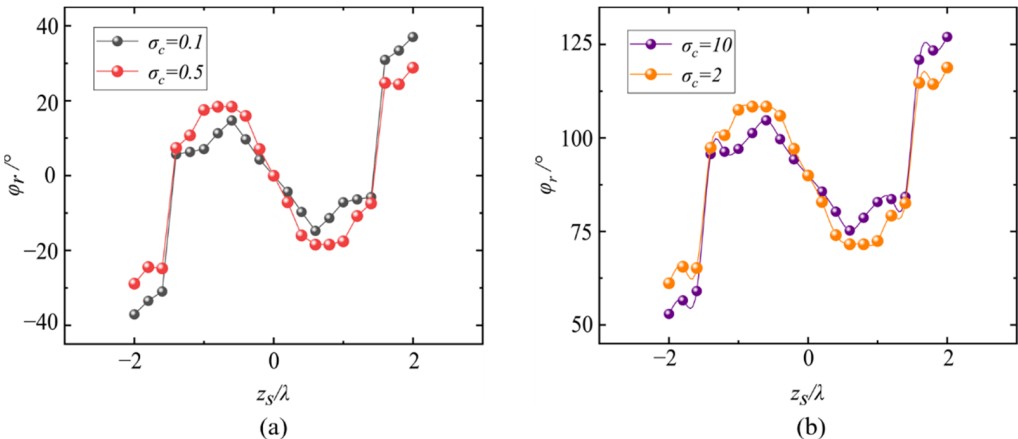

**Figure 9.** Same as Figure 8, but with more details in a narrower range ($-2\lambda < z_s < 2\lambda$).

Besides the effects of the propagation distance $z_s$ and the anisotropic parameter $\sigma_c$ on the transverse energy flow $\mathbf{p}_{xy}$, the beam order $l$, as we discussed for the longitudinal energy flow, will also influence the rotational behavior of $\mathbf{p}_{xy}$. Figure 10 depicts the distributions of the transverse energy $\left| \mathbf{p}_{xy} \right|$ along the propagation direction in the second-order case ($l = 2$). It can also be seen that, generally, the transverse energy also rotates in a counterclockwise manner as the beam propagates and, in Figure 11, the corresponding rotational angle $\varphi_r$ is shown, where the accumulated rotational angle $\int \varphi_r = 123.9°$ for $\sigma_c = 0.1, 10$ and $\int \varphi_r = 116.8°$ for $\sigma_c = 0.5, 2$ in the range $-14\lambda < z_s < 14\lambda$. We should note that abnormal rotational behavior also exists in the second-order case, which can be seen in Figure 12. It can be seen that this abnormal behavior is more complicated than it is in the case of $l = 1$. When $\sigma_c = 0.1, 10$, the clockwise rotation is in a very narrow range, about $-0.2\lambda < z_s < 0.2\lambda$; however, the accumulated rotational angle $\int \varphi_r$ is as big as $-70.0°$,

while, for $\sigma_c = 0.5, 2$, $\int \varphi_r$ is about $-18.4°$ within $-0.8\lambda < z_s < 0.8\lambda$. This abnormal rotational behavior occurs in the range closer to the focal plane, which may be caused by the redistribution of the topological structure of the transverse energy flow on account of the X-type vortex; this redistribution will be discussed in the following part.

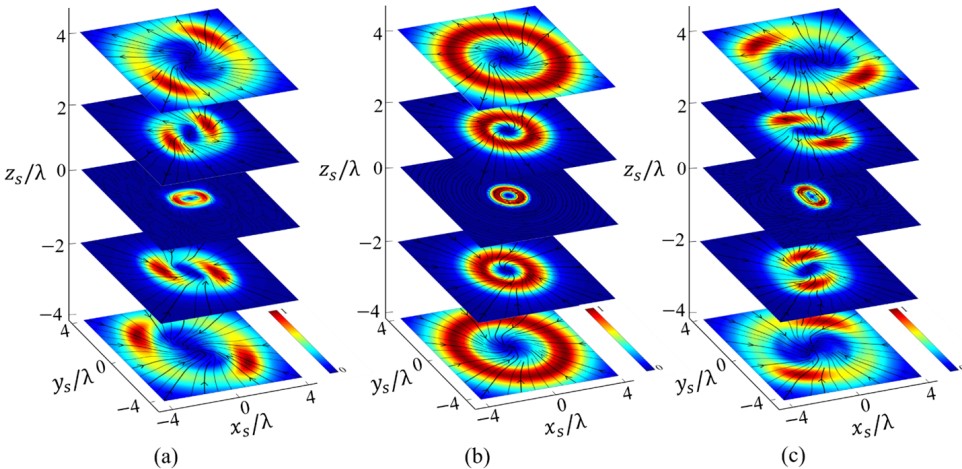

**Figure 10.** The transverse energy flow $\mathbf{p}_{xy}$ along the propagation direction: (**a**)$\sigma_c = 0.5$; (**b**)$\sigma_c = 1$; (**c**) $\sigma_c = 2$. In all the plots $l = 2$.

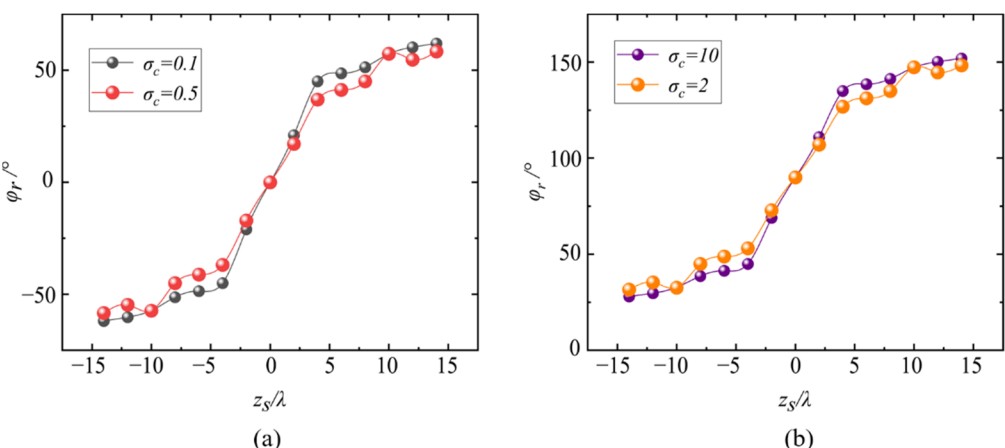

**Figure 11.** Variation in the rotational angle $\varphi_r$ for the transverse energy $\mid \mathbf{p}_{xy} \mid$ with the beam propagation. Here, (**a**) $\sigma_c = 0.1, 0.5$; (**b**) $\sigma_c = 2, 10$. In both plots $l = 2$.

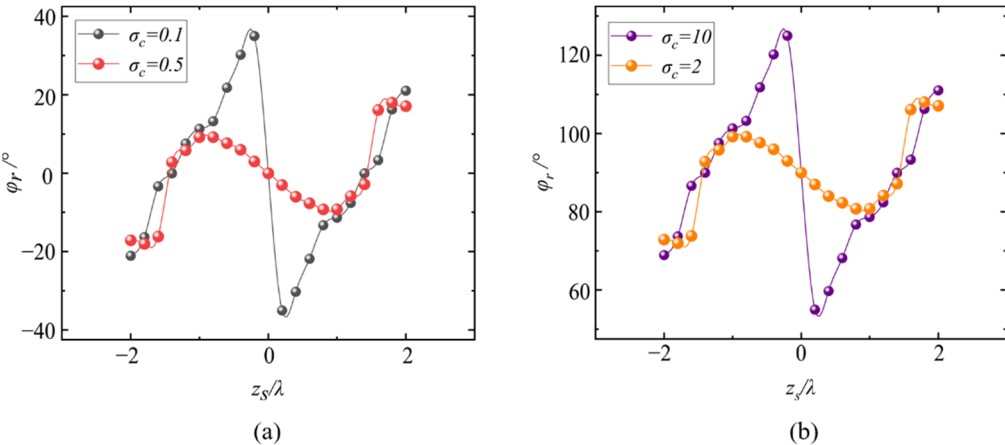

**Figure 12.** Same as Figure 11, but with more details in a narrower range ($-2\lambda < z_s < 2\lambda$).

In addition, here, we should note that the rotational angle $\varphi_r$ describes the rotational behavior of the transverse/longitudinal energy flow distribution (i.e., the energy flow 'pattern'), which is quite different from the 'skew angle' observed in traditional vortex beams [42,43]. The rotational behavior characterized by the 'skew angle' denotes the direction of rotation of the Poynting vector reflecting the orbital angular momentum effect of the vortex beams, while the rotation behavior discussed in this article is a kind of 'energy distribution' rotation essentially coming from the inconstant phase gradient of the X-type vortex, which, as we discussed, cannot be observed in a traditional (canonical) vortex beam. Furthermore, the energy flow in all of the above cases rotates in a counterclockwise manner due to the positive charge in the X-type vortex. If the topological charge of the X-type vortex is negative, i.e., $\sigma_c < 0$, and the charge is equal to $-l$ ($l \in \mathbb{N}$), the energy flow will rotate clockwise.

### 3.3. Transverse Energy Flow in the Focal Plane

The transverse energy flow $\mathbf{p}_{xy}$, including the flow lines (or the Poynting vectors) and the energy distribution, in virtue of the X-type vortex, also can exhibit interesting structures. Here, we will focus on the focal plane to examine these structures of the transverse energy flow.

First, we will look at the case of $l = 1$. Figure 13 illustrates the transverse energy flow on the focal plane for different values of the anisotropic parameter $\sigma_c$, where the energy strength is also denoted by color, and the black flow lines and the white arrows represent the energy flow lines and the transverse Poynting vectors, respectively. It can be found, on one hand, that, when $\sigma_c < 1$ (plots (a) and (b)) there are two energy maxima located on the $x_s$-axis, while, as $\sigma_c$ increases from 0.1, 0.5, 1, and 2 to 10, the energy maxima move from the $x_s$-axis to the $y_s$-axis, and, especially in the case of conventional vortex ($\sigma_c = 1$), the transverse energy is distributed uniformly along the azimuthal direction, i.e., there are no longer any maximum points. On the other hand, the flow lines for the conventional vortex ($\sigma_c = 1$) rotate around the beam center azimuthally and form circular shaped trajectories, while, when $\sigma_c \neq 1$, i.e., for the X-type vortex, the trajectories of these flow lines around the beam center become elliptical. More specifically, the energy flow lines near the outer sides of the maxima (such as the flow lines near $|x_s| > 0.8\lambda$ in plot (a), and the flow lines near $|y_s| > 0.8\lambda$ in plot €) derivate from the azimuthal trajectories around the beam center in the case of $\sigma_c \neq 1$, which implies that, although the topological charge of the energy flow in the beam center does not change in virtue of the X-type vortex for $l = 1$, the new Poynting singularities will be formed as $\sigma_c$ changes, i.e., the topological structure of the energy flow on the focal plane is changed. This redistribution of the topological structure is more obvious and typical as $l$ gets bigger, and, here, we adopt the case of $l = 2$ as an example for further analysis, which is depicted in Figure 14.

The transverse energy flow $\mathbf{p}_{xy}$ (with flow lines and vectors) in the focal plane with the anisotropic parameter changing from 0.1, 0.5, 1, and 2 to 10 for $l = 2$ is drawn in Figure 14. It can be seen that there exist three main singular points in the focal plane, the 'original' Poynting singularity at the beam center, 'O', and two (constructed) off-axis singularities 'A' and 'B'. When $\sigma_c = 0.1$ (plot (a)), points A and B are located on the $x_s$-axis with topological charge $+1$, while the point O has a charge of $-1$. As $\sigma_c$ increases to 0.5, the two off-axis singularities A and B move closer to the beam center, and, when $\sigma_c$ arrives at 1, the points A and B merge with original point O, resulting into a new singular point $O_+$ with topological charge $+1$, as shown in plot (c). This process obeys the conservation law of topological events. While, as $\sigma_c$ continues to increase, the singular point $O_+$ splits into three singularities again, the off-axis singularities A and B no longer exist on the $x_s$-axis; instead, they are located on the $y_s$-axis.

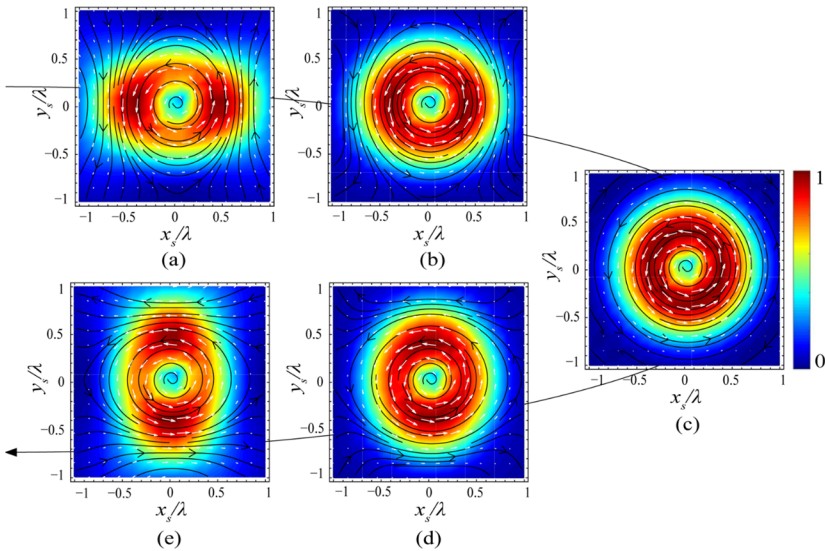

**Figure 13.** The transverse energy flow in the focal plane for $l = 1$: (**a**) $\sigma_c = 0.1$; (**b**) $\sigma_c = 0.5$; (**c**) $\sigma_c = 1$; (**d**) $\sigma_c = 2$; (**e**) $\sigma_c = 10$.

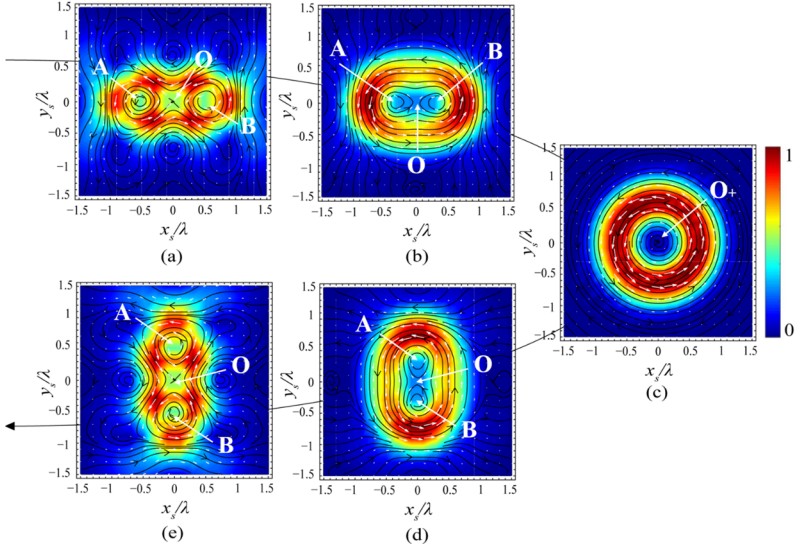

**Figure 14.** The transverse energy flow on the focal plane for $l = 2$: (**a**) $\sigma_c = 0.1$; (**b**) $\sigma_c = 0.5$; (**c**) $\sigma_c = 1$; (**d**) $\sigma_c = 2$; (**e**) $\sigma_c = 10$.

This topological event can be observed more obviously in a phase-type figure. Here, we define a complex transverse Poynting field $p_{xy}^{(c)}$ as:

$$p_{xy}^{(c)} = p_x + \mathrm{i}p_y, \tag{16}$$

where $p_x$ and $p_y$ are still the $x$- and $y$-Poynting components, respectively. By applying the equivalence of the vector field and its corresponding complex field, the topological structure of the Poynting vector field can also be described by the phase-structure of $p_{xy}^{(c)}$. Here, the phase singularity of $p_{xy}^{(c)}$ is equivalent to the vector singularity of the Poynting vector $\mathbf{p}_{xy}$, and, also, the topological charge of this phase singularity is equal to the charge of the vector singularity. The contour plot of the phase of the complex transverse Poynting field, $\arg\left[p_{xy}^{(c)}\right]$ (here $\arg[\cdot]$ means the argument or the phase of $p_{xy}^{(c)}$), is illustrated in Figure 15, where the intersections of different contours indicate the phase singularities of $p_{xy}^{(c)}$, which also means the Poynting (vector) singularity of the transverse energy flow $\mathbf{p}_{xy}$. In order to observe the topological behavior more clearly, the plots with $\sigma_c = 0.3$ and $\sigma_c = 3$

are added. The three main singular points A, B, and O are marked out in this figure, and it is easy to see that, as $\sigma_c$ increases, the points A and B, both with charge $+1$, move from the $x_s$-axis to merge with point O with charge $-1$, then to be created again on the $y_s$-axis. In addition to these three singularities and their topological reaction, we can also find other singularities in Figure 15, i.e., the points at the contour intersections around points A, B, and O, namely 'surrounding singularities.' As $\sigma_c$ increases (Figure 15a–d), it can be seen that these surrounding singularities will gradually annihilate each other and disappear when $\sigma_c = 1$ (plot (d)), and, as $\sigma_c$ continues to increase (Figure 15e–g), these surrounding singularities emerge again with their positions having a 90° rotation. More specifically, it is found that, for $\sigma_c = 1$, there is no surrounding singularity; instead, there exists a 'singularity ring,' i.e., an edge-type singularity (denoted in white in Figure 15d). This means that the multi vortex-type singularities of the energy flow appearing in the case of the X-type vortex will degenerate into a simple edge-type singularity for a canonical vortex. Thus far, the topological structures and the related reactions in the focal plane have been observed and analyzed. It is found that the topological structures change greatly with the topological reaction on account of the X-type vortex. Consequently, the transverse energy and flow directions are re-distributed, which implies that, by adjusting the anisotropic parameter $\sigma_c$, one can realize the manipulation of the transverse energy flows in the focal plane. Since the particle will move along the energy flow direction, the multi-vortex-type singularities of the transverse energy flow may provide a method to trap/rotate particles in the focal plane and transport them from one axis to another. It is well-known that the energy distribution strongly depends on the singular points; therefore, the topological reactions also mean that the complicated topological structures and related the transverse energy distribution in the transverse planes move nearer to the focus, which may be the reason for the abnormal rotation discussed in previous section.

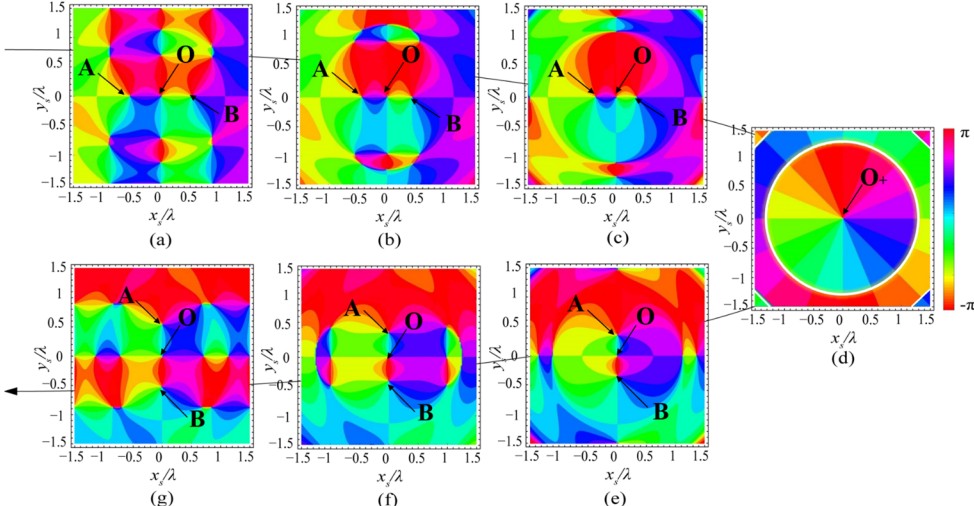

**Figure 15.** Contours of the phase of complex transverse Poynting field, $\arg\left[p_{xy}^{(c)}\right]$ on the focal plane for $l = 2$: (**a**) $\sigma_c = 0.1$; (**b**) $\sigma_c = 0.3$; (**c**) $\sigma_c = 0$; (**d**) $\sigma_c = 1$; (**e**) $\sigma_c = 2$; (**f**) $\sigma_c = 3$; (**g**) $\sigma_c = 10$.

## 4. Conclusions

In this article, the energy flow in a 3D vector field constructed by strongly focusing azimuthally-polarized beams with an X-type vortex is studied. It is found that, in virtue of the X-type vortex, the energy flow, including the longitudinal component and the transverse component, can rotate along the propagation direction. By adjusting the anisotropic parameter $\sigma_c$ and the vortex order $l$ of the X-type vortex, the location of the energy maxima and the rotation angle can be manipulated. Different from the longitudinal energy flow, the transverse energy flow will rotate inversely (i.e., in a clockwise manner) in a very short propagation distance near the focus, which may be caused by the complicated topological structures in that range. The transverse energy flow in the focal plane is also discussed,

and it is found that the energy distribution can show very distinguishing patterns as the anisotropic parameter $\sigma_c$ varies. Through defining a complex transverse Poynting field and applying the equivalence principle in singular optics, the topological behaviors of the transverse energy flow are analyzed, which shows that, instead of the simple edge-type singularity existing in the canonical vortex case ($\sigma_c = 1$), many vortex-type singularities emerge in the X-type vortex case, and their locations and topological reactions are determined by the anisotropic parameter $\sigma_c$. This research not only explores the physical properties of the X-type vortex, but also provides a method to construct a rotating energy flow with beam propagating and to form tunable energy flows in the focal plane, which may have applications in optical manipulations, such as rotating particles along the longitudinal or transverse directions, and in controlling the chirality of nanostructures [44,45].

**Author Contributions:** Conceptualization, X.P.; software, H.Z.; validation, H.Z. and T.Z.; formal analysis, X.P. and X.Z.; investigation, X.P. and H.Z.; writing—original draft preparation, H.Z. and T.Z.; writing—review and editing, X.P. and H.Z.; visualization, H.Z., T.Z. and X.Z. All authors have read and agreed to the published version of the manuscript.

**Funding:** This research was funded by the National Natural Science Foundation of China (NSFC) (11974281, 12104283), and by the Fundamental Research Funds for the Central Universities (No. GK202103021).

**Institutional Review Board Statement:** Not applicable.

**Informed Consent Statement:** Not applicable.

**Data Availability Statement:** The data presented in this study are available on request from the corresponding author.

**Conflicts of Interest:** The authors declare no conflict of interest.

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
