# Peer review of "Manipulation of Energy Flow with X-Type Vortex"

_photonics, doi:10.3390/photonics9120998_

Round 1
Reviewer 1 Report
In this paper, a new method for manipulating energy flow in a 3D vector field is proposed by using an azimuthally polarized beam with a X-type vortex in a high numerical aperture system. This research is supported by the convincing numerical simulations, and the corresponding analyses are detailed. This research is original, significant, and interesting. Also, this manuscript is beneficial to the audience and is well written. Therefore, it can be accepted for publication after addressing the following minor revisions:
1. The scale of the longitudinal and the transverse energy flow should be further refined in Figures.
2. Figure 3 is too simple and can be deleted, and it can be replaced with a text narrative.
Author Response
See the word file uploaded.

Reviewer 2 Report
This study investigates the spatial propagation of an X-type vortex, which is a conventional (L=n, δc=0) optical vortex with the characteristic anisotropic polarisation.
It focuses on a form of optical vortex that has not been well studied, and is considered to have significant research impact.
The reviewer would like to accept, but recommends some revisions.
1. It is a very interesting characteristic light, but how can it be obtained experimentally?
Can the authors provide the reader with information on how to use liquid crystal spatial light modulators, Q-Plates, etc.?
2. From Figures 4 and 8 and elsewhere, there appears to be no difference in the direction of flow depending on the size of δc and 1.
Is this direction determined by the topological charge?
3. Although only applications to optical tweezers are mentioned in the Conclusion, interesting effects may also be expected in ablation processes.
For example, elliptical needle fabrication or chiral needles twisted in various directions.
The following additional references are recommended for mention.
https://pubs.acs.org/doi/10.1021/nl301347j
https://www.sciencedirect.com/science/article/pii/S0169433218332069?via%3Dihub
Author Response
See the word file uploaded.
